

# Spatiotemporal Assimilation/Interpolation of Discharge Records through Inverse Streamflow Routing

Colby K. Fisher[1], Ming Pan[1], Eric F. Wood[1]

[1]Department of Civil and Environmental Engineering, Princeton University, Princeton, NJ USA

*Correspondence to*: Colby K. Fisher (ckf@princeton.edu)

**Abstract.** Poorly monitored river flows in many regions of the world have been hindering our ability to accurately estimate global water usage as well as the budgets and variability of the global water cycle. In-situ gauging sites, as well as a number of satellite-based systems, make observations of river discharge throughout the globe; however, these observations are often sparse due to, e.g., the sampling frequencies of sensors or a lack of reporting. Recently, efforts have been made to

develop methods to integrate these discrete observations to gain a better understanding of the underlying processes. This paper presents an application of a fixed interval Kalman smoother based model, called Inverse Streamflow Routing (ISR), to generate spatially and temporally continuous river discharge fields from discrete observations. The method propagates the observed information across all reachable parts of the river network (up/downstream from gauging point) and all reachable times (before/after observation time) using a two-sweep procedure that first propagates information backward in time to the furthest

upstream locations (inverse routing) and then propagates it forward in time to the furthest downstream locations (forward routing). The key advantages of this approach are that it (1) maintains all the physical consistencies embodied by a diffusive wave routing model (flow confluence relationships on the river network and the resulting mass balance, wave velocity and diffusivity), (2) updates the lateral influx (runoff) at pixel level (furthest upstream) to guarantee exhaustive propagation of observed information, (3) works both with a first guess of initial river discharge conditions from a routing model (assimilation)

and without a first guess (pure interpolation of observations). Two sets of experiments are carried out under idealized conditions as well as real-world conditions provided by U.S. Geological Survey observations. Results show that the method can effectively reproduce the spatial and temporal dynamics of river discharge in each of the experiments presented. The performance is driven by the density of the gauge network as well as the quality of the data being assimilated. We find that when assimilating the actual USGS observations, the performance decreases relative to our idealized scenario; however, we

are still able to produce an improved discharge product at each validation site. With further testing as well as global application, ISR may prove to be a useful method for extending our current network of global river discharge observations.

## 1 Introduction

In the application of water resources for human use, as well as the monitoring and prediction of global hydrologic hazards, such as floods and droughts, a comprehensive understanding of globally distributed runoff and river discharge is extremely



important. In many regions of the world, river flows are poorly monitored with in-situ observations and the collection of the available observations for consumption by global end-users has proven to be a difficult challenge, as evidenced by the available records from the Global Runoff Data Center (GRDC) (Fekete et al., 2012). Streamflow records are typically most complete in the relatively developed and populous parts of the world; however, streamflow data in many regions are often considered

proprietary, resulting in, among other issues, difficult problems in the management of water resources in transboundary rivers (see e.g., Biancamaria et al., 2011; Pavelsky et al., 2014).

Besides the usefulness of global near real time river discharge data for water management, there is a great need for observations of river discharge data to further our understanding of the global water cycle and its representation by reanalysis and climate models. While other observational sources for the terrestrial water budget have become more readily available

from satellite remote sensing, the lack of comprehensive river discharge observations has resulted in a key flux (runoff) in climate models and large-scale land surface models being poorly constrained by observations over much of the global land surface (Sahoo et al., 2011). Furthermore, the amount of water stored at the land surface and its space-time variability are poorly known. To better serve the global hydrologic community, there is then a need for methods which can make further use of the currently available global discharge data sources.

These data sources can be divided into the following sets: 1) observations based on in situ measurements (gauges); 2) estimates based on remotely sensed observations (e.g. satellite altimetry, synthetic aperture radar); and 3) estimates based on land surface models (LSMs) and routing models (Pan and Wood, 2006). Traditionally, the data provided by sets 1 and 2 can be thought of as point observations of river discharge along a river network, whereas set 3 can provide us with a spatially distributed representation of discharge throughout our basins of interest, derived from modelled runoff fields. These two

variables can be connected by the process of streamflow routing, where the spatially distributed runoff generated at the land surface flows over the hillslope and through a river network to become streamflow in the river channels. From this process we can then say that the streamflow (as measured at specific points in space and time) is the integrated response to the runoff through a subset of time and space. Due to this process, all studies using streamflow as representative of basin runoff are limited to applications where the temporal differences can be ignored or accounted for (e.g. for long time scale studies the

aggregation of the runoff data allows us to ignore the temporal differences) (Sahoo et al., 2011; Sheffield et al., 2009; Pan et al., 2012). There is then a need for new methods that are able to derive spatially and temporally continuous records of runoff and river discharge from the available data sources.

The goal of this study is to present an application of a methodology by which we can integrate and use the point scale observations of river discharge to derive a product that is spatially and temporally continuous. One possible method is the

combination of point observations with spatially distributed model estimates through assimilation. Due to the integrated nature of the streamflow generation process, any discharge assimilation must be able to propagate information throughout the range of influence in a basin for any given gauge. Additionally, we must be able to assimilate all available observations in a basin simultaneously in time and space, to resolve conflicts due to observational errors. There have been a number of recent studies to investigate the potential for such discharge assimilation using a wide variety of methods, e.g. (Andreadis et al., 2007;



Biancamaria et al., 2011; Paiva et al., 2013; Pan and Wood, 2013). While these methods are often robust and comprehensive, the large computational burden, in particular for those using Ensemble Kalman Filters such as that used by Andreadis et al. (2007), limits their potential for rapid global application. In addition, many of these methods simply adjust discharge in a forward sense and do not fully account for the upstream spatial and temporal correlations of the streamflow generation process.

Alternative methods have focused on the use of kriging based statistical techniques to derive spatially distributed estimates of river discharge, e.g. (Paiva et al., 2015 and Yoon et al., 2013). These statistical methods often show good agreement for the reconstructed discharge with little computational cost but are highly dependent on the formulation of the covariance matrix for each river system. Here we propose the use of an assimilation and interpolation scheme for creating spatially complete and temporally continuous river discharge records from point observations based on the Inverse Streamflow Routing (ISR) model,

which was previously used by Pan and Wood (2013) for the generation of spatially distributed runoff fields to be used in land surface modelling applications, such as the calibration of model parameters. The new approach maintains the important structure of the streamflow generation process with a relatively low computational burden and guarantees an exhaustive propagation of observed information to all reachable locations across the river network and reachable times. The new approach can be more effective compared to the assimilation and interpolation methods discussed previously, which perform an

assimilation by adjusting the state variables (e.g. water height/volume, flow rate, etc.) and propagate the observed information much less exhaustively.

## 2 Methods

In short, the proposed method tries to propagate the observed discharge information across all reachable parts of the river network (up/downstream from gauging point) and all reachable times (before/after observation time) using a two-sweep

procedure that first propagates information backward in time to the furthest possible upstream (inverse routing) and then propagates it forward in time to the furthest possible downstream (forward routing). Figure 1 provides a detailed illustration of the proposed scheme. The first sweep of the procedure, known as the Inverse Streamflow Routing (ISR), developed by Pan and Wood (2013) to generate spatially distributed runoff fields, plays the key role here (left side of Fig. 1). The ISR helps to guarantee an exhaustive propagation of observed information by updating the boundary influx (runoff) at pixel level (the

furthest possible upstream) throughout the entire spatial and temporal domains. The second sweep simply re-runs the same routing model forward using the runoff fields derived from the first sweep to reconstruct continuous discharge values everywhere (right side of Fig. 1). Since ISR does not require an initial guess of discharge from the routing model (Pan and Wood, 2013), the proposed method works for both data assimilation (if an initial guess exists) and pure interpolation of observations (without an initial guess). Since the discharge records are ultimately created by a routing model, this approach

preserves all the physical consistencies embodied by the chosen routing model and its parameters such as the flow confluence relationship on the river network and the resulting mass balance, wave velocity and diffusivity (if a diffusive wave routing model is used). Such a strong physical consistency can hardly be implemented by methods based on statistical correlations





between different gauging points or different state variables in the routing model, for example, the river kriging method (Paiva et al., 2013). When used as an interpolator, the proposed method can also exactly reproduce the input observations at gauging locations/times (Pan and Wood, 2013). The mathematical formulation of this method is described below.

## 2.1 Routing Model Formulation

5   The basic routing model selected for this work is the University of Washington (UW) routing model (Lohmann et al., 1996; Nijssen et al., 2001), which provides a simple linear routing scheme that is commonly coupled with LSMs. This model routes runoff through two processes. The first of these is the drainage of the runoff water within a grid cell to the outlet of the grid cell as governed by a known unit hydrograph function (UHF). This is given by equation ( 1 ) below, where $u(t)$ is the UHF, $r(t)$ is the pixel runoff, and $o(t)$ is the pixel outflow.

$$o(t) = \int_0^t r(t-r)u(\tau)d\tau \tag{1}$$

10  The second process then governs the travel of water in channels between pixels through the one-dimensional diffusive wave equation. This is given by equation ( 2 ), where $q$ is the streamflow generated by the pixel outflow at a distance $x$ downstream, $C$ is the channel wave velocity and $D$ is diffusivity.

$$\frac{dq}{dt} = D\frac{\delta^2 q}{\delta x^2} - C\frac{\delta q}{\delta x} \tag{2}$$

This model is linear as long as the parameters $C$ and $D$ are assumed not to be a function of the streamflow, i.e., retention effects such as lakes and reservoirs as well as human management are not considered, and thus it is a good candidate for our inversion.

15  These two stages of the routing process are then solved together using the form presented by equation ( 3 ) below, where $i(x,t)$ is the impulse response function as defined by equation ( 4 ).

$$q(x,t) = \int_0^t r(t-\tau)u(t-\tau)i(x,t)d\tau \tag{3}$$

$$i(x,t) = \frac{x}{2t\sqrt{\pi t D}}exp\left\{-\frac{(Ct-x)^2}{4Dt}\right\} \tag{4}$$

By integrating equation ( 3 ) for all upstream pixels, denoted as $all(g)$, for any given gauge $g$ at discretized time steps, we can determine the streamflow at any gauge location, $Q(g,t)$ as shown in equation ( 5 ) .





$$Q(g,t) = \sum_{all(g)} q(x,t) \tag{5}$$

This formulation serves as the basic routing model for inversion and for the final reconstruction of discharge from the inverted runoff fields.

**2.2 Inverse Streamflow Routing Model**

Using the routing model presented above, the fixed interval Kalman smoother can now be established for the inversion process

following Pan and Wood (2013). First, the routing model must be written in a linear state space form as a function of input states as seen in equation ( 6 ).

$$y_t = H_0 x_t + H_1 x_{t-1} + \cdots + H_k x_{t-k} + \epsilon_t \tag{6}$$

In this form, $y_t$ is a vector of the discharges at a number of gauges in the basin and $x_t$ is a matrix of the runoff for all cells at time $t$. Because of the integration to determine flow at each gauge, the model requires runoff information up to a lag time of $k + 1$ steps, which is the travel time of the basin. Finally, $H_t$ represents the measurement operator matrix, whose elements

represent the amount of runoff that one specific cell will contribute to each gauge at a given time. These values are calculated from the impulse response function. As a result of the integration described above, there is a need for the solution of this inverse problem for multiple time steps at once, which gives rise to the fixed interval smoothing component of this inversion. Through a time augmentation, the model can ultimately be written in the Kalman filter form as shown in equation ( 7 ) below (Pan and Wood, 2010).

$$\hat{x}''_t = \hat{x}'_t + K_t(y'_t - H'\hat{x}'_t - L'\hat{x}'_{t-k}) \tag{7}$$

In this form, $\hat{x}'_t$ is the initial guess of the time augmented runoff fields, $y'_t$ is the time augmented streamflow measurements, $H'$ and $L'$ are time augmented measurement operators, $K_t$ is the Kalman gain as given by equation ( 8 ) and $\hat{x}''_t$ is the updated estimate of the runoff fields.

$$K_t = P_t H'^T (H' P_t H'^T + R_t)^{-1} \tag{8}$$

The Kalman gain represents a weighting of the update to the runoff fields and is controlled by $P_t$, which represents the error covariance matrix of the initial forecast for the runoff, and $R_t$, which is the error covariance matrix of the gauge measurements.

For this study, we perform a set of idealized experiments in which we set $R_t$ equal to 0, such that the inversion process provides a maximum correction to the initial runoff guess. The error covariance ($P_t$) is defined as a diagonal matrix of the long-term mean runoff error variance. In practical applications this error term will be derived from the error utilized in the particular



form of the discharge observations. It should be noted that this method can function without an estimate of the initial runoff conditions (a null field) and thus, it also works for streamflow interpolation in which river discharge is reconstructed purely from observations. With the first ISR completed through ISR, the second sweep of flow reconstruction is done by running the same routing model in a forward sense with the new runoff influxes.

## 2.3 Experimental Design and Study Area

For this study we perform two sets of streamflow interpolation experiments over the Ohio River basin. The Ohio River basin, along with the Tennessee River in the southern part of the basin, is a large basin covering an area of approximately 490000 km². This basin contains a wide variety of river sizes that drain a mix of developed, undeveloped and agricultural areas, all of which are monitored by the United States Geological Survey (USGS) with a dense network of gauges. This monitoring network makes the basin a good candidate for these streamflow interpolation experiments, as we will be able to use the extensive USGS observations as another set of data inputs.

The first of these experiments performs the inversion using synthetically created streamflow values as a proof of concept for the method. The goal of this experiment is to see if the streamflow interpolation method can generate the true discharge, given varying levels of information about the prior runoff conditions in the basin. The second experiment is the same as the previous, except that the synthetic gauge data is replaced with actual USGS gauge data. In this experiment, the performance of the ISR method is evaluated under "real world" conditions given that the routing model does not account for the effects of human management and will produce streamflows that are likely different from the observed streamflows. Flow charts of these two experiment sets can be seen in Fig. 2. Each of these experiments were run for the entire year of 2009. This period was selected because the daily discharge characteristics were representative of the climatology, with some individual high flow events. Based on the previous work of Pan and Wood (2013), the wave velocity parameter and the smoothing window for the Ohio River basin were set at 1.4 m/s and 70 days, respectively.

## 2.4 Data

In each of the experiments, the NLDAS 0.125 degree meteorological dataset (Cosgrove et al., 2003) is used to force the Variable Infiltration Capacity (VIC) LSM (Liang et al., 1994, 1996) to produce runoff fields that are considered the "true" runoff. The NLDAS precipitation forcings were chosen for this experiment as they combine hourly radar analyses and daily gauge observations and are considered to provide a comprehensive and reliable set of forcings over the United States (Pan et al., 2010). This NLDAS derived runoff is then used with the routing model described previously to generate synthetic streamflow values at set evaluation sites ("pseudo gauges") for the study period. These 75 sites are the routing model grid cells in which an actual USGS gauge is located. 25 of these gauge sites are designated as validation sites and the remaining 50 sites provide river discharge time series to be assimilated in the ISR model. The selection of these gauge sites was based solely on finding gauges within the basin that had relatively complete discharge records (>95% days available) for the year 2009 and the distribution of validation sites was random. The use of these USGS gauge based sites for validation of the synthetic model





results also allows for later experiments and comparisons with the actual USGS observations. The distribution of these pseudo gauge stations as well as a representation of the routing model river basin can be seen in Fig. 3.

The generated synthetic streamflows are considered the "true" observations and are used in the streamflow interpolation process to correct an initial estimate of river discharge (derived from an initial estimate of daily runoff that is

also routed using the Lohmann routing model). To investigate the impact of this initial runoff estimate, we perform each synthetic experiment with three daily initial conditions. These are: (1) a long term mean value of runoff applied over the entire basin (same value in every grid cell for every day in the study period), (2) a daily climatology of runoff at each grid cell, derived from the NLDAS forced VIC LSM, and (3) daily runoff values derived from the VIC LSM forced with the real-time TRMM Multi-Satellite Precipitation Analysis (TMPA) version 3B42RT (Huffman et al., 2007) precipitation product. The

TMPA product was selected for this experiment as it is globally available between 60° N and 60° S at a 3-hour temporal and 0.25° spatial resolution. The product was interpolated to 0.125° to force the VIC simulations (Pan et al., 2010). While this product is not as accurate as the ground observation based NLDAS product, it is globally available and can be used along with the VIC LSM to provide us with a realistic initial forecast of runoff even when ground observations do not exist (Pan et al., 2010). The results of these three purely synthetic experiments and three USGS observation-based experiments are presented

in Section 3.

## 3 Results

Following the above methodology, six discharge interpolation (reconstruction) experiments were performed. To evaluate the performance of the interpolation in each of these experiments we compute the Nash-Sutcliffe Efficiency (NSE) at each of the 25 pseudo gauges designated for validation in Fig. 3. The NSE is a measure of model performance and is defined in equation

(9), where $Q_o$ is the mean of observed discharges, $Q_m^t$ is modeled discharge at time t, and $Q_o^t$ is observed discharge at time t (Nash and Sutcliffe, 1970).

$$NSE = \frac{\sum_{t=1}^{T}(Q_m^t - Q_o^t)^2}{\sum_{t=1}^{T}(Q_o^t - \overline{Q_o})^2}$$

( 9 )

The NSE may range from -∞ to 1, with an efficiency of 1 meaning that there is a perfect match between the modeled discharge and the observations (or the synthetic truth). An efficiency of 0 indicates that the model is just as accurate as the mean of the observations and a value less than 0 indicates that the mean would be a better predictor than the model.

### 3.1 Synthetic Discharge Interpolation

For the first set of experiments we follow the procedure outlined by Fig. 2a. An example of the discharge interpolation can be seen in Fig. 4, where the time series of discharge are shown for 2 of the 25 validation gauges. The runoff initial conditions for this set of reconstructions was the climatological daily runoff. Figure 4a, which represents a downstream gauge with a large



upstream area, shows good performance for the ISR method. We find that the NSE increased from 0.527 to 0.995, indicating a large increase in the model performance through assimilation. By examining the overall time series, we can see that the assimilation was able to correct for a majority of the conditions imposed by the initial guess of runoff. For example, between days 50 and 100 we can see that the initial guess had significantly higher flows compared to the synthetic truth, where these

high flows centered around day 50, and the assimilation was able to reconstruct this quite well. Figure 4b shows the same results for an upstream gauge with a smaller contributing area, where we observe an increase in NSE from 0.049 to 0.986 after the discharge reconstruction. Similar to the previous example, this performance is quite good, indicating that the ISR methodology can be effective for reconstructing spatially and temporally continuous discharge records. Despite this, we find that for some gauges with the smallest upstream areas, which potentially contain less assimilated gauges than others, the

reconstructions will occasionally miss the temporal dynamics of the synthetic truth, such as between days 150 and 200 in Fig. 4b.

        Figure 5 illustrates the evaluation of the NSE values across all of the validation sites for each of the three initial runoff conditions. By plotting the distribution of NSE values in the validation gauge set for the initial guess and the reconstructed discharge we can see the performance improvement from the streamflow interpolation method. Across all of the

initial conditions we can see that there is an increase in performance for many of the gauges, with a shift in the NSE values towards 1. In particular, we find that the null initial guess of runoff performs the best (Fig. 5a). This is likely because we are not imposing any temporal or spatial dynamics on the runoff, just a mean value, which allows for the interpolation to adequately reconstruct the temporal dynamics of the synthetic truth. In contrast to this, we found that the experiment with initial conditions based on the TMPA observed precipitation performed the worst, as there were often differences in when events such as high

flows started or the magnitude of these events, which the assimilation was not able to completely correct for. Despite these differences, we find that the ISR method is able to do a good job of creating discharge records across all initial conditions, with a noticeable increase in performance in each case.

        To further illustrate the impact of upstream area and gauge density on the performance of the interpolation, we plot the upstream area of each validation gauge versus the NSE for reconstructed discharge in Fig. 6. For each experiment we can

see the same pattern, with a wide variety of NSE values for gauges with upstream areas less that $10^4$ km$^2$ while basins larger than this have NSE values consistently between 0.9 and 1. These larger sub-basins incorporate the information of other upstream gauges assimilated, allowing for a more accurate reconstruction of discharge. In addition to this, the integrative nature of the routing and smoothing procedure dampens many of the short high flow events, allowing the larger sub-basins to exhibit consistently better performance given reliable upstream observations.

**3.2 USGS Gauge Interpolation**

In addition to these purely synthetic experiments, we evaluated the performance of the streamflow interpolations under real world conditions by substituting daily observed USGS river discharge values for the synthetic truth used previously. Here we present the results of these three USGS based experiments, varying the initial runoff conditions in the same manner as the





previous experiments. Figure 7 illustrates the performance of the ISR model for discharge reconstruction when assimilating these in situ river discharge observations. Again, we find that the method works well for the two evaluation gauges presented, with the larger basin (Fig. 7a) improving the NSE from 0.166 to 0.862 and the smaller basin (Fig. 7b) improving from -0.061 to 0.942. Comparing these results to those from the purely synthetic experiment presented in Fig. 4, we see that the use of the

USGS data degrades the performance of the reconstruction. This is likely due to the non-linear components of flow, such as reservoirs, dams or backwater effects, which are present in this basin and can significantly alter the flow from what this linear routing model predicts. Additionally, during some of the peak flow periods (such as days 100 to 150 in Fig. 7b), we can see instances where the reconstructed discharge is greater than the synthetic truth. This is a result of a numerical correction done in the model where physically unrealistic negative runoff values resulting from each Kalman smoother update are reset to a

value of zero. The effect of this correction is more apparent in the assimilation of these USGS observations than in the synthetic experiments.

Figure 8 presents the overall results of these experiments, again displaying the distributions of NSE values resulting from the initial guess and the discharge reconstruction. For all initial conditions, the ISR model is able to create some improvement in the reconstructed discharge values; however, the degree of improvement is noticeably less. In contrast to the

synthetic experiments, the initial guess of runoff derived from the TMPA precipitation resulted in the best performance for the interpolation while the null and climatological initial guesses performed similarly, exhibiting a smaller shift in the NSE values for all the evaluation gauges. In part, this is due to the non-linear flow characteristics in the USGS observations that we are not representing, as there are often conflicting estimates of the spatial distribution of discharge between the USGS observations and the TMPA precipitation-based discharge, which lead to a decrease in the innovation term provided by the Kalman

smoother. Another potential cause of this decreased performance could be errors in the river discharge observations themselves. In-situ observations are likely to have errors of varying magnitudes; however, we are treating these observations as error free for the purposes of model evaluation. As a result, any potential errors in these observations will then be transferred to errors in the final reconstructed discharge estimates.

## 4 Discussion

The need for global discharge and runoff observations and estimates is not new and there have been a number of recent studies that have taken different approaches to generating spatially and temporally distributed discharge from point observations (Andreadis et al., 2007; Biancamaria et al., 2011; Paiva et al., 2013; Paiva et al., 2015). The ISR method is an alternative approach to these methods, which allows for the creation of spatially distributed discharge fields that are not only spatially consistent but are also consistent through time, due to the application of a Kalman Smoother. The results of these experiments

have shown that the ISR method can produce a good representation of discharge throughout a basin river network given a wide variety of initial conditions. In particular, the interpolation from USGS observations is promising, as we are able to generate a very close representation of the discharge conditions throughout the basin with little to no prior information about the specific



distribution of runoff present. This indicates that the ISR method may be able to extend the usefulness of observations in basins with sparse gauge networks, such as many underdeveloped regions of the world. It is also important to note that the ISR method produces fields of runoff that are consistent with the observed discharges, which may prove beneficial for the calibration and optimization of land surface model processes in poorly gauged basins.

5        Although these experiments have illustrated the potential for the ISR method to be used for river discharge interpolation in global basins, it is important to acknowledge that these experiments are idealized and thus, do not contain all of the potential errors and uncertainties that would be present in a real-world application. As discussed previously, our method is limited by the lack of a non-linear routing model, the presence of error free observations and the overall parameterization of the routing model (static parameters for the wave velocity and the diffusivity). With regards to errors in the observations, Pan

and Wood (2013) tested the impact of these errors on the ISR models' ability to reconstruct runoff fields and found that these errors could potentially be significant enough to remove any positive improvements from the assimilation procedure. In real world applications we will need to carefully consider the error characteristics of the data sources to be assimilated, as these will have a significant impact on the quality of the final discharge product.

       Another important variable in the performance of this method is the availability and selection of gauges (or pseudo

gauges in synthetic experiments) for assimilation and evaluation. In this study we present the results of assimilating one specific configuration of available gauging sites as illustrated in Fig. 3. To understand better how the results of these experiments might change if the network of gauges were configured differently, we performed a sensitivity study by generating 100 random configurations of gauges to assimilate and evaluate from the total set of 75. These gauge networks were then used to reconstruct discharge in each of the previous 6 experiments, evaluating the NSE at each of the 25 evaluation sites, for each

possible network configuration. The results of these simulations are illustrated in Fig. 9, where the gauge configurations for each experiment are ranked according to the median NSE of reconstructed discharge. In addition to this, the box and whisker plots illustrate the spread of performance for each configuration, as well as any potential outliers. Finally, the yellow box in each experiment represents the specific simulation results that are presented in this paper.

       Focusing first on the results of the synthetic experiments with a null initial condition of runoff (Fig. 9a), we can see

that there is a significant amount of variation in the specific distributions of NSE; however, there is little change in the median value or the lower limit of NSE values across all configurations. This indicates that regardless of the network configuration chosen, we are able to reasonably reconstruct spatially and temporally distributed discharge within the basin river network. Looking across the three initial conditions for the synthetic experiments (Fig. 9a, c, e) shows results comparable to those presented previously, with the null and climatological initial runoff conditions providing relatively similar performance. The

TMPA derived initial conditions show a distribution of median NSEs that is slightly lower than the prior experiments. It is also interesting to note that with increasing information in the initial conditions, the spread of the model performance increases considerably. This is a further illustration of the case where large differences between the initial guess and true conditions can degrade the effectiveness of the ISR model for generating discharge throughout an entire basin.



Finally, the results of these 100 random configurations for the experiments using in-situ USGS discharge observations are shown in Fig. 8b, d, and f. Overall, we can observe the pattern of decreasing performance from null to TMPA derived initial conditions is still present. Across all three experiments, the range of median NSE values is larger than that for the synthetic experiments, indicating that the selection of gauges for assimilation in this real-world scenario has a more significant impact.

We also find that the spread of the NSE distributions is greater than those in the synthetic experiments, further reinforcing the influence of the previously discussed error and uncertainty sources in the ISR method. Understanding and constraining these errors will be critical to future applications of ISR.

## 5 Conclusion

In this study we have developed a two-sweep method for reconstructing spatially and temporally continuous discharge records

from discrete observations of discharge, in which the first sweep applies the ISR method (Pan and Wood, 2013) to propagate observed information backward in time/space and the second sweep re-runs the same routing model to propagate information forward in time/space. The new formulation is expected to offer more complete propagation of observed information in time and space (thus a better performance) and a better physical consistency than existing approaches. The core algorithm of this method is formulated as a Kalman Smoother, allowing for an assimilation/interpolation of discharge from all available

observations of discharge in a basin. By assimilating and validating against synthetic and real observations at 75 gauging sites in the Ohio river basin, the new approach has illustrated good performance across all experiments. In particular, the results of the discharge reconstructions given a null initial runoff condition are promising, as they illustrate the ability of the streamflow assimilation/interpolation methodology to create continuous discharge records in a basin where we do not have a good climatology or a calibrated hydrologic model.

The performance of this method will be limited by the availability and quality of gauge data, the specific initial conditions chosen, the parameterization of the routing model and the exclusion of non-linear features such as dams (Yin et al, 2016a; Yin et al., 2016b). Further work is needed to determine how this method will perform as the density of the gauge network is reduced or as the amount of days missing from a gauge's discharge record is increased, as would be the case in many of the global basins which do not currently have robust observation networks. Temporally sparse observations are

particularly challenging for this type of assimilation/interpolation, as specific extreme events could be missed entirely, or the method may not have enough data to maintain a correction through time from the initial guess. At a minimum, ISR can be used to reconstruct a distributed representation of discharge from one or a few in-situ gauge observations; however, the more information that can be provided for the assimilation, the more likely we are to produce an accurate estimate of the discharge conditions in that basin.

To improve upon the density of observations in sparsely gauged regions, this methodology could be extended to perform interpolations from remotely sensed river discharge products, such as those from current generation satellite altimetry, or the upcoming NASA Surface Water and Ocean Topography (SWOT) mission (Alsdorf and Lettenmaier, 2003; Durand et



al., 2010; Pavelsky et al., 2014). The SWOT mission, scheduled to launch in 2021 is of particular interest, as this will be a swath altimeter designed to provide global observations of water surface elevation and slope, from which river discharge can be estimated. Within the 21-day repeat cycle, a river reach will be observed 2-4 times, on average (Biancamaria et al., 2010). The prospects for such a space borne sensor are great, especially with respect to the global coverage; however, due to the inclination of the orbit these observations are not evenly distributed in time or space and thus they will not be as complete as the USGS observations used here. In general, we believe that this form of streamflow interpolation using the ISR method could serve as a framework for creating spatially and temporally continuous discharge records from sparse observations like the future SWOT mission. Careful consideration will be required to account for the gaps in observations and the unique error characteristics of these remotely sensed discharge observations.

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





**Figure 1: Two-sweep procedure for spatio-temporal assimilation/interpolation of discharge records. The first sweep (lower left) propagates observed information collected at gauging points upstream and backward in time following the Inverse Streamflow Routing method developed in Pan and Wood (2013) and derives continuous runoff fields (lateral influx at furthest possible upstream). The second sweep (lower right) propagates information downstream and forward in time (regular routing) to create continuous discharge values everywhere. The stacked spatial maps at the top illustrate how the observed information at a single point in time/space propagates backward in time/space (upper left) and how discharge is reconstructed from the integrated runoff (upper right).**



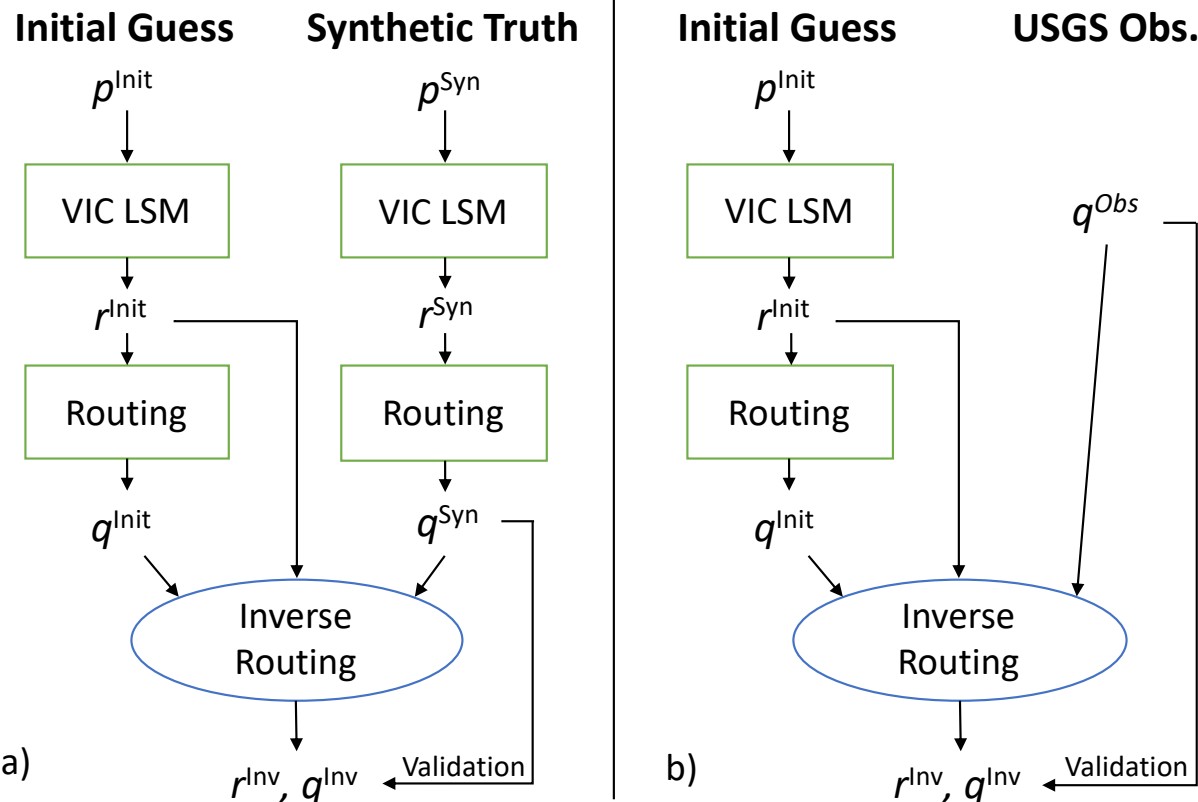

**Figure 2: a) Overall process flow diagram for the synthetic experiments with the ISR model, where p is the input precipitation for the VIC LSM distributed over the study domain and study period, r is the runoff fields distributed over the same space, and q is discharge at discrete points during the study period. The superscripts "Init" and "Syn" represent the initial guess and the synthetic truth, respectively, while the "Inv" superscript indicates the products resulting from the model. b) Overall process flow diagram for the ISR model, substituting actual USGS discharge observations for the synthetic truth of the previous experiments.**





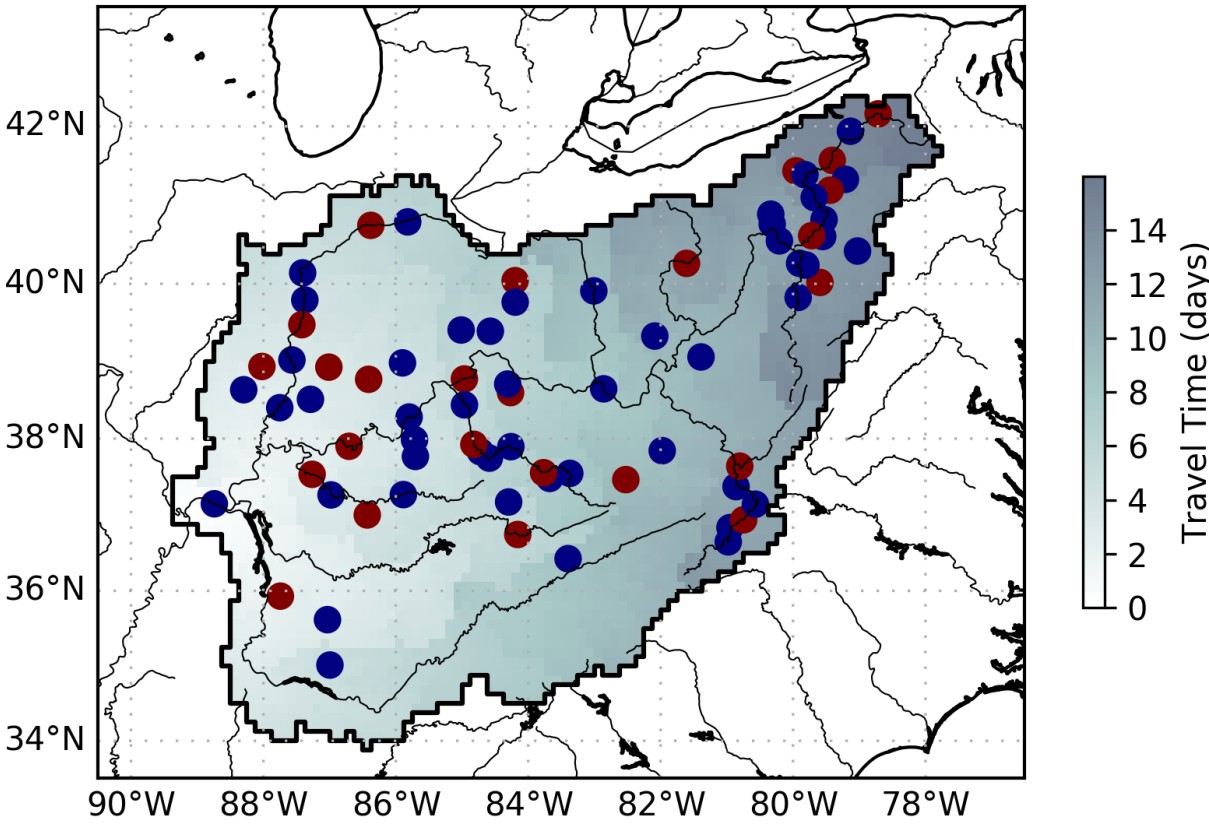

**Figure 3: The Ohio River basin modelled at 0.125 degree resolution and the distribution of the 75 USGS gauge sites used in the creation of pseudo gauges for assimilation. Blue dots represent those gauges used in the assimilation and interpolation while the red dots represent those gauges which will be reconstructed for evaluation. The background shading indicates the travel time from each grid cell to the outlet of the basin.**




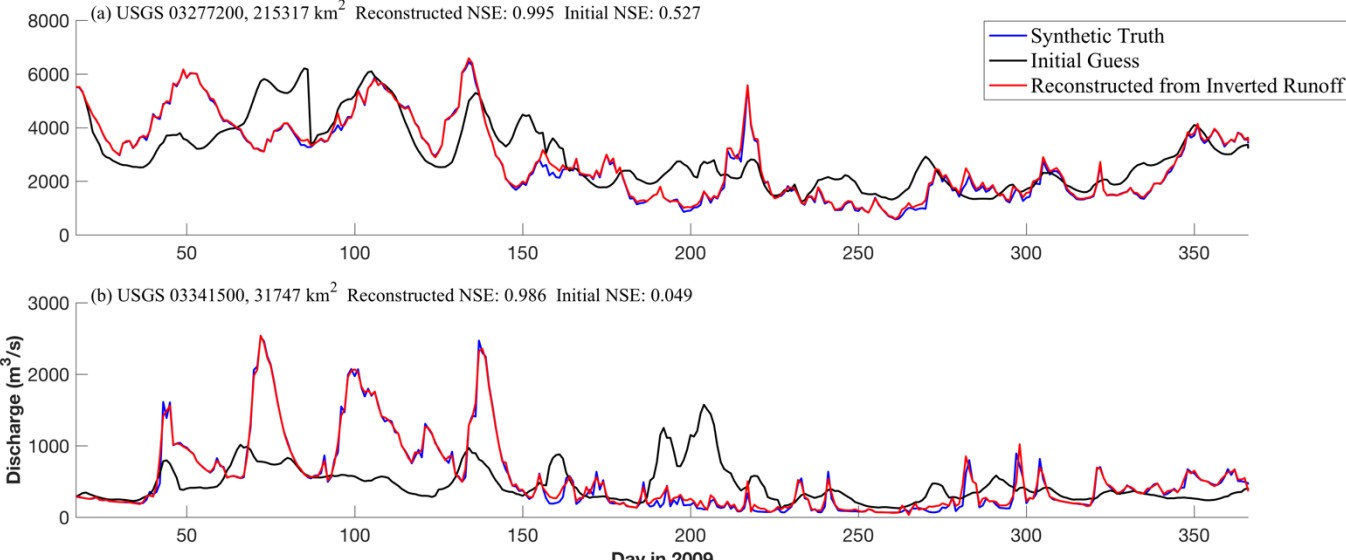

**Figure 4: Reconstructed discharge time series during 2009 for two of the 25 evaluation sites when the ISR model was run using the climatological initial guess of runoff conditions. The blue line represents the synthetic truth discharge used in the inverse routing, the black line illustrates the discharge derived from our initial guess of runoff, and the red line illustrates the reconstructed discharge. NSE values are given for the initial guess and the reconstruction in relation to the synthetic truth.**





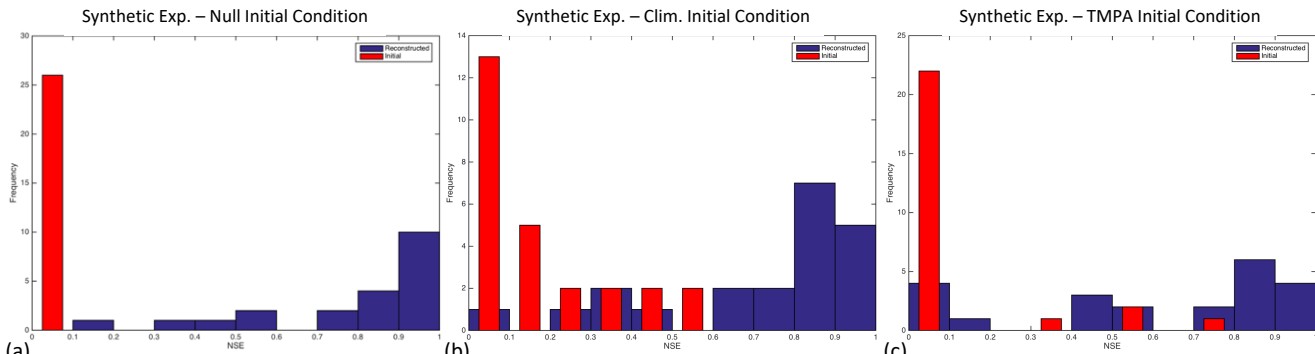

**Figure 5: Distributions of NSE values for the three synthetic experiments with varied initial conditions of runoff. These daily initial conditions are: a) Null (uniform mean runoff over the entire basin), b) Climatology (average daily runoff over the entire period from NLDAS), and c) TMPA (runoff derived from TMPA precipitation and VIC LSM). In each plot, the red bars illustrate the distribution of NSE values for discharge generated from the initial guess of runoff and the blue bars indicate the same distribution after reconstruction with the inverse routing method.**





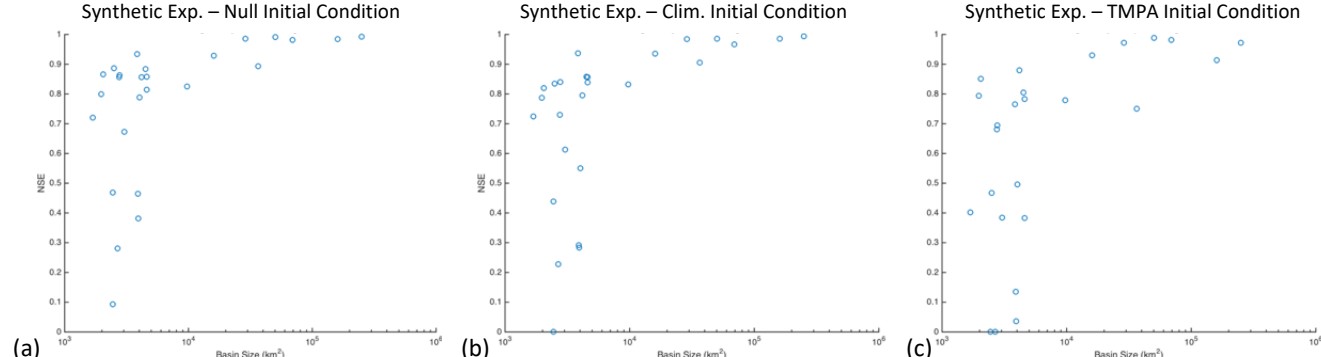

**Figure 6: Distribution of NSE values for each of the evaluation sites versus the size of the upstream area for each gauge. The ordering and experiment names are the same as those in Figure 5.**



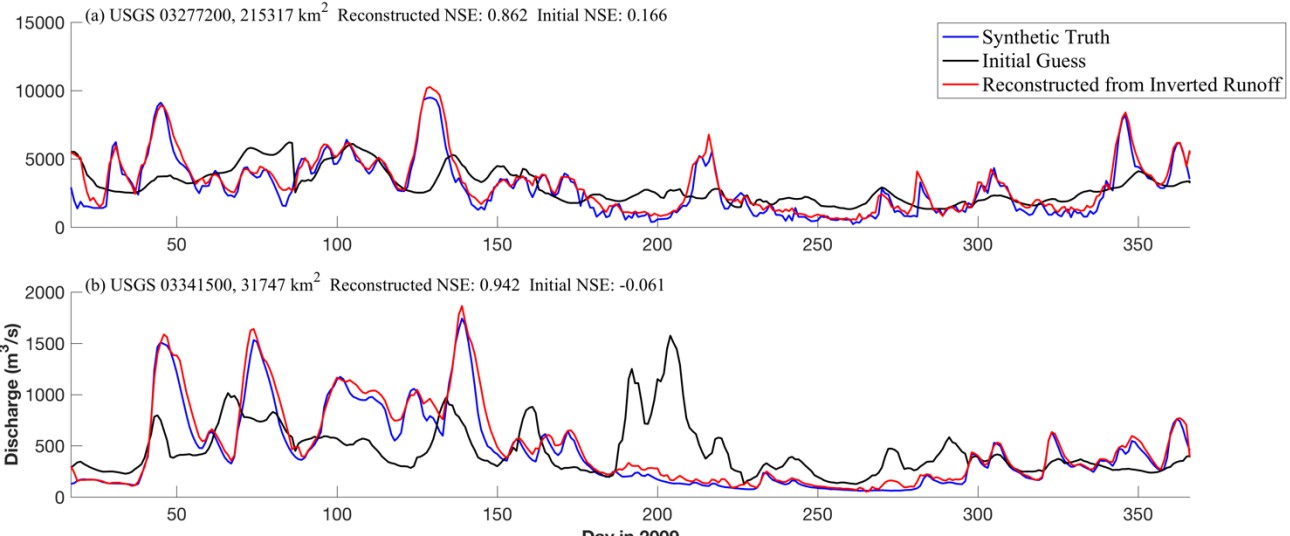

**Figure 7: Reconstructed discharge time series during 2009 for two of the 25 evaluation sites when the ISR model was run using the climatological initial guess of runoff conditions. Here the discharge data assimilated and compared against (the "truth") are USGS observations.**





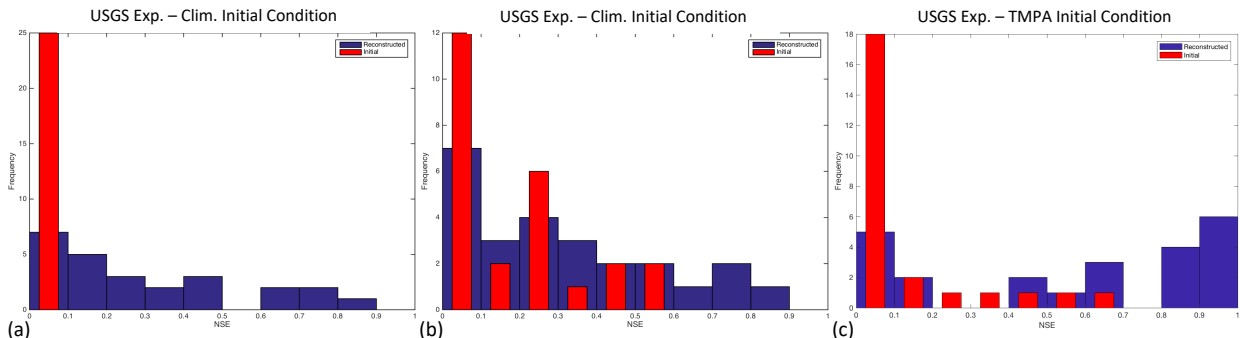

**Figure 8: Distributions of NSE values for the three synthetic experiments with varied initial conditions of runoff and USGS observations as the synthetic truth. The ordering is the same as that in Figure 5.**



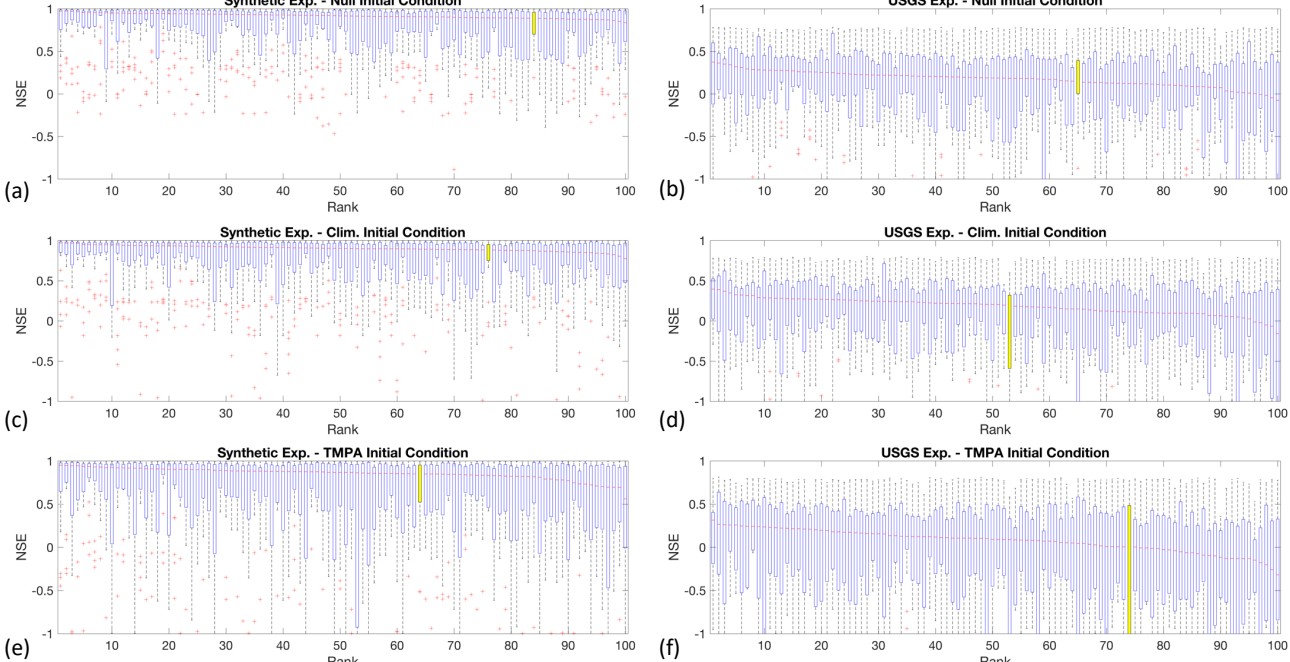

**Figure 9: Box plots of NSE values at 25 validation sites for 100 random configurations of the gauge network. Experiments are dived into two categories: the entirely synthetic discharge reconstructions (a,c,e) and the discharge reconstructions from USGS observations (b,d,f). These experiments are further differentiated by the three initial runoff conditions used: Null (a,b), Climatology (c,d), and TMPA derived (e,f). The yellow box plots represent the gauge network configuration used for the results presented in this study.**