# Peer review of "Spatiotemporal Assimilation/Interpolation of Discharge Records through Inverse Streamflow Routing"

_Hydrology and Earth System Sciences, 2018_

## Referee Comment (RC1) · L. Read (Referee) · 7 Jun 2018

The purpose of the paper is to demonstrate a methodology for calculating 'continuous' streamflow from a network of point observations, with the targeted application to estimate global discharge in areas with sparse measurements. The topic and methodology are of great interest to the community as we are making progress on increasing access to observations worldwide and growing the remote-sensing based methods for estimating a host of hydrologic variables including river discharge. I have several main concerns regarding the manuscript and then list several minor ones for improvement:

Major: My primary concern is on the generalizability of this approach in other basins

with differing characteristics. Of course the method can be applied, but the performance seems highly unknown. For example, given that there are differences in performance of the method based on drainage area, I can imagine other characteristics - mean annual flow, degree of landuse/urbanization, management, etc., also leading to differences in performance. In my opinion since the method was established in a previous work (2013), this paper should include some sensitivity (besides the distribution of gauges, which was a very nice experiment) analysis, either using real data (preferred) or synthetic. Further, only one year (2009) is tested - why? Was 2009 particularly wet or dry? These could also impact the results.

The second major question is the relative performance of the statistical (kriging) method compared with the one presented. While I agree with the authors that a statistically based approach sacrifices mechanistic understanding, it is important to understand the context of where this work fits in and how it compares (as the authors mention in the introduction).

The third major point is on the application itself, where this case was presented for the Ohio river basin, which is a heavily gauged basin (making it great for an initial test, but perhaps poor for demonstrating how the method could be used for global discharge). In my opinion, the likely errors and missing data in the observations are a big concern. In the US we have a relatively low occurrence of missing obs compared with other places in the world where I imagine this could be applied, so the question is: at what point does this method break-down - what are the stress-test results (how many missing obs lead to performance worse than taking the mean or some other metric at the nearest gauge, or applying a statistical method)?

Minor: -Only NSE was used as a metric of performance, but there are other statistics that would be interesting depending on whether the user is interested in floods/droughs - 7Q10, bias, etc. -Suggest revising the language in the results that often repeats "we can see" (pg 8). -Avoid saying whether the method did a "good job". Allow the reader to determine that from the results. -Fig 5: make axes and legend font larger -Fig 9:

caption has a typo. Dived –> Divided?

---

## Referee Comment (RC2) · Anonymous Referee #2 · 12 Jul 2018

My main concern is in regards to the transferability of this method to the remote basins with sparse data, extensively discussed in the conclusion. Given the experiments run, I'm not convinced this method would do any better than the simpler statistical methods typically used for ungagged basins. Line 18-21, page 6, you write "Based on the previous work of Pan and Wood (2013), the wave velocity parameter and the smoothing window for the Ohio River basin were set at 1.4 m/s and 70 days, respectively." In that previous work, Pan and Wood set the values based on a streamflow model calibrated to the gages. How would you get these parameters without having already had a good working model with a lot of data in the area? In my experience, models are very sensitive to wave celerity values, which vary with discharge in non-linear models. So to

leave a value constant through time, especially with no model/data to back it up and also considering interannual variability: this all seems quite risky. I feel this paper could be greatly improved by backing down the conclusions and instead perhaps alluding to the possibility that such conclusions will be tested in future work somehow (at least comparing the method to other methods in such basins). Or, as the previous reviewer suggested, more sensitivity studies should be presented, in regards to the velocities assumed and the period of record tested on, since the Pan and Wood (2013) paper has already presented the novel method of inverse routing.

1. Does the paper address relevant scientific questions within the scope of HESS? Yes.

2. Does the paper present novel concepts, ideas, tools, or data? Somewhat. This paper is an extension of the ideas in the 2013 paper by Pan and Wood, that was published in HESS. That paper was much more novel. I find this paper interesting, but mostly in the concept that was already introduced in Pan and Wood. I am not convinced that the claim of the method being better that traditional methods in remote basins without good data is an accurate claim, so I'm not convinced of the novelty of the idea extension.

3. Are substantial conclusions reached? Yes, but I'm not sure they are justified.

4. Are the scientific methods and assumptions valid and clearly outlined? Yes, this paper is extremely well written.

5. Are the results sufficient to support the interpretations and conclusions? The interpretations are supported but not the conclusions.

6. Is the description of experiments and calculations sufficiently complete and precise to allow their reproduction by fellow scientists (traceability of results)? Yes.

7. Do the authors give proper credit to related work and clearly indicate their own new/original contribution? Yes.

8. Does the title clearly reflect the contents of the paper? Yes.

9. Does the abstract provide a concise and complete summary? Yes.

10. Is the overall presentation well structured and clear? Yes.

11. Is the language fluent and precise? Yes.

12. Are mathematical formulae, symbols, abbreviations, and units correctly defined and used? Yes.

13. Should any parts of the paper (text, formulae, figures, tables) be clarified, reduced, combined, or eliminated? No.

14. Are the number and quality of references appropriate? Yes.

15. Is the amount and quality of supplementary material appropriate? Yes.

---

## Author Response (AR1)

**Responses to Reviewer Comments**

Dear HESS editor:

Here we want to express our sincere appreciation to you, reviewer Dr. Read, and the anonymous reviewer for your great help with our work. The comments, criticisms, suggestions, insights and guidance provided to us are invaluable to improving our work. Following these comments, we have carefully addressed the issues raised, added additional analysis (e.g. extra performance metrics and sensitivity tests), and revised the manuscript accordingly.

Besides textual changes, we have performed a number of additional experiments and analysis, following the suggestions from the reviewers. These additional experiments and analysis include, primarily:

1. Repeat the discharge interpolation experiments with different wave velocity (celerity) values and assess the sensitivity on the interpolation skills;
2. Repeat the discharge interpolation experiments with different numbers of gauges remove and assess the sensitivity;
3. Calculate the Kling-Gupta Efficiency (KGE) score and its three components (bias ratio, Pearson correlation, and relative variability) in order to better understand how and why the interpolation can help improve the discharge estimation;

In addition, we also want to thank both you and the two reviewers for your great patience and continued support for our work despite the long delay in revisions due to personal difficulties.

The reviewer comments are quoted in normal **black** font, the responses are highlighted in **blue** color, and the text from the revised manuscript is cited in **red** color.

Sincerely,

Colby K. Fisher, Ming Pan, and Eric F. Wood

**Reviewer 1:**

The purpose of the paper is to demonstrate a methodology for calculating 'continuous' streamflow from a network of point observations, with the targeted application to estimate global discharge in areas with sparse measurements. The topic and methodology are of great interest to the community as we are making progress on increasing access to observations worldwide and

growing the remote-sensing based methods for estimating a host of hydrologic variables including river discharge. I have several main concerns regarding the manuscript and then list several minor ones for improvement:

Major: My primary concern is on the generalizability of this approach in other basins with differing characteristics. Of course the method can be applied, but the performance seems highly unknown. For example, given that there are differences in performance of the method based on drainage area, I can imagine other characteristics - mean annual flow, degree of landuse/urbanization, management, etc., also leading to differences in performance. In my opinion since the method was established in a previous work (2013), this paper should include some sensitivity (besides the distribution of gauges, which was a very nice experiment) analysis, either using real data (preferred) or synthetic.

Yes, we agree that sensitivity analysis is necessary to better understand the generalizability of the proposed approach. In addition to gauge distributions, we have added the sensitivity tests against the wave velocity (a parameter that is hard to estimate/calibrate without observations) and gauge density (number of available gauges to remove in the flow reconstruction).

A new figure, Figure 9, (along with additional analysis in lines 21-31 P10, text copied below) has been added to show the results of two sensitivity experiments:

P10 Lines 21-31:

"While the work of Pan and Wood (2013) focused on the inversion of spatial runoff fields, the ISR method is used to generate integrated discharge values; therefore, it is important to see how sensitive the model will be to similar errors. Figure 9 presents the results of two sensitivity experiments: 1) selection of a different velocity parameter, and 2) decreasing amounts of available gauge data. Both of these experiments were carried out using the null initial conditions of runoff. For the velocity parameter, the calibrated velocity of 1.4 m/s produces the best performance; however, there is still skill in the discharge reconstruction for velocity values from 50% to 200% of the optimal value (Fig. 9a). With regards to potentially limited availability of observations, the number of gauges assimilated in the ISR model was decreased from 0% to 50% of the full set by randomly removing gauges (Fig. 9b). In all cases, the KGE values indicate adequate model performance; however, these results are dependent on the information contained within each observation, as removing a gauge with a larger contributing area is likely to have a larger impact on the overall model performance that a smaller one."

[Figure]

[Figure]

Figure 9: Box plots of KGE values at 25 validation sites for two sensitivity experiments: a) varying wave velocity parameters, b) removing gauges from the observation set. In each of these experiments the null initial condition was used. The mean of each set is denoted with a red X.

Further sensitivity analysis against different flow climatology, geolocations, landcover/landuse, water management, etc. are also necessary. Given that the primary goal of this study is on the introduction of methodology and proof of concept, we think such larger scale experiments should be carried out in a separate and more application-focused study. So we included such experiments in a second paper that is in review:

Fisher, C.K., M. Pan, and E. F. Wood, 2019: Deriving Continuous Discharge Records from Future SWOT Observations. Water Resources Research, in review.

Further, only one year (2009) is tested - why? Was 2009 particularly wet or dry? These could also impact the results.

The goal of this paper was to perform experiments to illustrate the feasibility of a numerical method to reproduce given discharge records. Year 2009 is an "average" and "typical" year for the study area (Ohio basin) with a normal seasonal cycle and flow values that cover most of the high and low ranges of regional hydrology. There were no dramatic/extended droughts or floods in 2009.  Such a choice can minimize the impact of many compounding factors like the deficiencies of simple diffusive wave routing under extreme conditions (high flow and low flow). Also experiments and observations over a "typical" period can be more generalizable.

This information was added into the text in lines 16-18 P6:

"Each of these experiments were run for the entire year of 2009. This period was selected because the daily discharge characteristics were representative of the climatology, with some individual high flow events but no dramatic/extended droughts or floods. Such a choice can minimize the impact of many compounding factors like the deficiencies of simple diffusive wave

routing under extreme conditions (high flow and low flow). In addition, experiments and observations over a "typical" period can be more generalizable."

The second major question is the relative performance of the statistical (kriging) method compared with the one presented. While I agree with the authors that a statistically based approach sacrifices mechanistic understanding, it is important to understand the context of where this work fits in and how it compares (as the authors mention in the introduction).

We agree with the reviewer that it is important for this work to be considered in the context of the previous work done on statistical river kriging (e.g. Paiva et al. (2015) and Yoon et al. (2013)). A careful assessment of the existing statistical approaches shows that it is very difficult to fairly reproduce the kriging results without significant effort from the original authors, given that (1) the study area, study period, training data, underlying parameters etc. are all different, (2) the kriging approach is much more complicated than spatial kriging and its efficacy will depend on training/tuning. Quantitative comparisons against the statistical methods are necessary and we think it is much more fair and convincing to conduct such comparative studies together with all relevant researchers in this area and have all participants working on the same study area/period and input data.

That said, we fully acknowledge that we do not have quantitative metrics to accurately prove the better efficacy of this method. Adjustments to the discussions/conclusions (P12, L28-31) are made to stress this point. Despite this, we believe that the fact that this model accounts for a physical representation of the river system (in the form of a river routing mode, albeit a simple one) is important. We believe that this difference alone provides strong evidence that there should be better physical consistency than with a purely statistical method.

P12, L23-31:

"This work should be considered in the context of the previous work done on statistical river kriging, e.g. Paiva et al. (2015) and Yoon et al. (2013). However, due to various limitations (e.g., lack of properly trained kriging parameters over the study area), no experiments are carried out to compare the performance of ISR based approach to statistical river kriging. We do not have quantitative metrics to prove the incremental improvement made by our method."

The third major point is on the application itself, where this case was presented for the Ohio river basin, which is a heavily gauged basin (making it great for an initial test, but perhaps poor for demonstrating how the method could be used for global discharge). In my opinion, the likely errors and missing data in the observations are a big concern. In the US we have a relatively low occurrence of missing obs compared with other places in the world where I imagine this could be

applied, so the question is: at what point does this method break-down - what are the stress-test results (how many missing obs lead to performance worse than taking the mean or some other metric at the nearest gauge, or applying a statistical method)?

We thank the reviewer for the thoughts on the application of this method to basins with larger errors and missing observations. One way to assess the applicability over other basins and under other conditions is to perform sensitivity tests (as suggested earlier) - see the results from the sensitivity experiments in the response to the earlier comment.

These results are presented in lines 21-31 P10:

"While the work of Pan and Wood (2013) focused on the inversion of spatial runoff fields, the ISR method is used to generate integrated discharge values; therefore, it is important to see how sensitive the model will be to similar errors. Figure 9 presents the results of two sensitivity experiments: 1) selection of a different velocity parameter, and 2) decreasing amounts of available gauge data. Both of these experiments were carried out using the null initial conditions of runoff. For the velocity parameter, the calibrated velocity of 1.4 m/s produces the best performance; however, there is still skill in the discharge reconstruction for velocity values from 50% to 200% of the optimal value (Fig. 9a). With regards to potentially limited availability of observations, the number of gauges assimilated in the ISR model was decreased from 0% to 50% of the full set by randomly removing gauges (Fig. 9b). In all cases, the KGE values indicate adequate model performance; however, these results are dependent on the information contained within each observation, as removing a gauge with a larger contributing area is likely to have a larger impact on the overall model performance that a smaller one."

Also, the work presented here was done in the context of the need for global river discharge products for the upcoming NASA Surface Water and Ocean Topography NASA mission. A thorough study of the applicability of this method to observations that are sparse in both space and time has actually been done and is under review in Water Resources Research. In that study the ISR method is applied to synthetic SWOT observations and investigates the performance of the method given the varying discharge estimation errors and intermittent observations of the mission:

Fisher, C.K., M. Pan, and E. F. Wood, 2019: Deriving Continuous Discharge Records from Future SWOT Observations. Water Resources Research, in review.

Minor:
-Only NSE was used as a metric of performance, but there are other statistics that would be interesting depending on whether the user is interested in floods/droughts - 7Q10, bias, etc.

On top of NSE, we have now added more metrics of skill including the Kling-Gupta Efficiency (KGE) score and its three subcomponents: bias ratio (a measure of bias relative to mean), correlation coefficient (a measure of linear correlation), and relative variability (a measure of scaling bias in dynamic range). These new metrics provide a thorough breakdown on how the interpolation may or may not improve the initial guess. Most flood and drought metrics like 7Q10 are very sensitive to extreme values and the values calculated from the data are less robust due to the relatively limited experimental period. So they are not implemented here. Figures 5 and 8 have been updated to include five metrics (NSE, KGE and three KGE breakdowns):

[Figure]

Figure 5: Distributions of NSE values (ab,c), KGE values (d,e,f) and its component statistics: bias ratio β (g,h,i), correlation coefficient r (j,k,l), and relative variability α (m,n,o),for the three synthetic experiments with varied initial conditions of runoff (shown in three columns). These

daily initial conditions are: 1) Null (uniform mean runoff over the entire basin), 2) Climatology (average daily runoff over the entire period from NLDAS), and 3) TMPA (runoff derived from TMPA precipitation and VIC LSM). In each plot, the red bars illustrate the distribution of the statistic values for discharge generated from the initial guess of runoff and the blue bars indicate the same distribution after reconstruction with the inverse routing method.

[Figure]

Figure 8: Distributions of NSE and KGE values for the three synthetic experiments with varied initial conditions of runoff and USGS observations as the synthetic truth. The ordering is the same as that in Figure 5.

-Suggest revising the language in the results that often repeats "we can see" (pg 8).

These items have been revised throughout the manuscript to vary the language. For example:

P8L7-12:

"We find that the NSE increased from 0.527 to 0.995, indicating a large increase in the model performance through assimilation. By examining the overall time series, it is evident that the assimilation was able to correct for a majority of the conditions imposed by the initial guess of runoff. For example, between days 50 and 100 the initial guess had significantly higher flows compared to the synthetic truth, where these high flows centered around day 50, and the assimilation was able to reconstruct this quite well. Figure 4b shows the same results for an upstream gauge with a smaller contributing area, where we observe an increase in NSE from 0.049 to 0.986 after the discharge reconstruction."

P8L17-19:

"These plots of the distribution of KGE values in the validation gauge set for the initial guess and the reconstructed discharge illustrate the performance improvement from the streamflow interpolation method. Across all of the initial conditions there is an increase in performance for many of the gauges, with a shift in the KGE values towards 1."

P9L6:

"For each experiment the same pattern is observed, with a wide variety of NSE values for gauges with upstream areas less than $10^4 km^2$ while basins larger than this have NSE values consistently between 0.9 and 1."

P9L18-21:

"Comparing these results to those from the purely synthetic experiment presented in Fig. 4, it is clear that the use of the USGS data degrades the performance of the reconstruction. This is likely due to the non-linear components of flow, such as reservoirs, dams or backwater effects, which are present in this basin and can significantly alter the flow from what this linear routing model predicts. Additionally, during some of the peak flow periods (such as days 100 to 150 in Fig. 7b), there are instances where the reconstructed discharge is greater than the synthetic truth. "

-Avoid saying whether the method did a "good job". Allow the reader to determine that from the results.

Statements such as these have been edited throughout the paper to address this. For example:

P8L6:

"Figure 4a, which represents a downstream gauge with a large upstream area, shows the performance of the ISR method."

P9L2:

"Despite these differences, we find that the ISR method is able to create discharge records across all initial conditions, with a noticeable increase in performance in each case due to the improvement in correlation and variability."

P10L10:
"The results of these experiments have shown that the ISR method can produce a representation of discharge throughout a basin river network given a wide variety of initial conditions."

P12L5-8:
"By assimilating and validating against synthetic and real observations at 75 gauging sites in the Ohio river basin, the new approach has illustrated the potential for discharge reconstructions across all experiments. In particular, the results of the discharge reconstructions given a null initial runoff condition are promising, as they illustrate the ability of the streamflow assimilation/interpolation methodology to create continuous discharge records in a basin where we do not have an adequate climatology or a calibrated hydrologic model."

-Fig 5: make axes and legend font larger

The figure has been updated accordingly, following the change to KGE metrics.

-Fig 9: caption has a typo. Dived –> Divided?

This typo has been fixed (P24L2).

**Reviewer 2:**

My main concern is in regards to the transferability of this method to the remote basins with sparse data, extensively discussed in the conclusion. Given the experiments run, I'm not convinced this method would do any better than the simpler statistical methods typically used for ungagged basins. Line 18-21, page 6, you write "Based on the previous work of Pan and Wood (2013), the wave velocity parameter and the smoothing window for the Ohio River basin were set at 1.4 m/s and 70 days, respectively." In that previous work, Pan and Wood set the values based on a streamflow model calibrated to the gages. How would you get these parameters without having already had a good working model with a lot of data in the area? In my experience, models are very sensitive to wave celerity values, which vary with discharge in non-linear models. So to leave a value constant through time, especially with no model/data to back it up and also considering interannual variability: this all seems quite risky.

We thank the reviewer for the thoughts on the applicability of this model to data poor regions. We fully acknowledge that you need a good forward model for the ISR method to work appropriately. Without this it will be hard to accurately reproduce the discharge in complex basins. Note that the unknown wave celerity (velocity) problem over ungauged basins is not a challenge unique to this ISR based approach but a common challenge to all statistical approaches as well where the lag correlations need to come from somewhere. In Paiva et al. (2015), such lag correlations are based on the analysis of the same diffusive wave model with the need for the same celerity parameter. No method can get a free lunch on the ungauged basin parameter problem simply because we can't make it up, either explicitly in the ISR approach or inexplicitly in statistical approaches, if nothing has been observed. We fully recognize this issue and adjustments have been made to the discussion to stress this caveat. The added sensitivity analysis is presented in lines 21-31 p10 and Figure 9 (as shown in response to the first reviewer above):

P10 Lines 21-31:

"While the work of Pan and Wood (2013) focused on the inversion of spatial runoff fields, the ISR method is used to generate integrated discharge values; therefore, it is important to see how sensitive the model will be to similar errors. Figure 9 presents the results of two sensitivity experiments: 1) selection of a different velocity parameter, and 2) decreasing amounts of available gauge data. Both of these experiments were carried out using the null initial conditions of runoff. For the velocity parameter, the calibrated velocity of 1.4 m/s produces the best performance; however, there is still skill in the discharge reconstruction for velocity values from 50% to 200% of the optimal value (Fig. 9a). With regards to potentially limited availability of observations, the number of gauges assimilated in the ISR model was decreased from 0% to 50% of the full set by randomly removing gauges (Fig. 9b). In all cases, the KGE values indicate adequate model performance; however, these results are dependent on the information contained within each observation, as removing a gauge with a larger contributing area is likely to have a larger impact on the overall model performance that a smaller one."

[Figure]

[Figure]

Figure 9: Box plots of KGE values at 25 validation sites for two sensitivity experiments: a) varying wave velocity parameters, b) removing gauges from the observation set. In each of these experiments the null initial condition was used. The mean of each set is denoted with a red X.

At the same time, modeling communities have proposed different ways to "transfer" parameters from gauged basins to ungauged basins (parameter "transfer" or "regionalizations") according to various similarity measures and physical constraints. One common approach is to the so-called "parameter regionalization", i.e., to relate parameter values to various globally available hydrologic, climatic, and morphologic features like accumulation area, mean annual precipitation in contributing area, temperature, slope, channel planform geometry, local valley curvature, soil properties, etc. This approach trains a regression relationship between available features and parameters over gauged areas and then applies such a regression globally. More recently, global estimates of river width has been available, adding another powerful tool for parameter estimation:

Allen, G. H., & Pavelsky, T. M. (2018). Global extent of rivers and streams. Science, 361(6402), 585–588. https://doi.org/10.1126/science.aat0636

Allen, G. H., David, C. H., Andreadis, K. M., Hossain, F., & Famiglietti, J. S. (2018). Global Estimates of River Flow Wave Travel Times and Implications for Low-Latency Satellite Data. Geophysical Research Letters, 45(15), 7551–7560. https://doi.org/10.1029/2018GL077914

We have been working on the challenge of poor parameter availability and some of our progress has been summarized in our global river discharge modeling study here:

Lin, P., Pan, M., Beck, H., Yang, Y., Yamazaki, D., Frasson, R., David, C.H., Durand, M., Pavelsky, T.M., Allen, G.H., Gleason, C., Wood, E., 2019: Global reconstruction of naturalized river flows at 2.94 million reaches. Water Resources Research, doi: 10.1029/2019WR025287.

Also, the future SWOT satellite mission itself (a main motivation of this study) will provide river height/width/slope information over many ungauged basins, and the hope is that new data from SWOT may also help infer flow parameters (e.g. lag correlation analysis).

I feel this paper could be greatly improved by backing down the conclusions and instead perhaps alluding to the possibility that such conclusions will be tested in future work somehow (at least comparing the method to other methods in such basins). Or, as the previous reviewer suggested, more sensitivity studies should be presented, in regards to the velocities assumed and the period of record tested on, since the Pan and Wood (2013) paper has already presented the novel method of inverse routing.

Language regarding the performance of the model has been reduced in strength. For example, P8L6, P9L2, P10L10, P12L5-8 (see response to Reviewer 1 above).

P8L6:
"Figure 4a, which represents a downstream gauge with a large upstream area, shows the performance of the ISR method."

P9L2:
"Despite these differences, we find that the ISR method is able to create discharge records across all initial conditions, with a noticeable increase in performance in each case due to the improvement in correlation and variability."

P10L10:
"The results of these experiments have shown that the ISR method can produce a representation of discharge throughout a basin river network given a wide variety of initial conditions."

P12L5-8:
"By assimilating and validating against synthetic and real observations at 75 gauging sites in the Ohio river basin, the new approach has illustrated the potential for discharge reconstructions across all experiments. In particular, the results of the discharge reconstructions given a null initial runoff condition are promising, as they illustrate the ability of the streamflow assimilation/interpolation methodology to create continuous discharge records in a basin where we do not have an adequate climatology or a calibrated hydrologic model."

In addition, we believe that the content of the last two paragraphs of the manuscript allude to the future work which will test the applicability of this method to discharge interpolation when discharge observation errors and missing observations are incorporated. Along with this, we believe the focus of this manuscript is different than the work presented by Pan and Wood (2013), where the previous work focused on runoff, versus discharge as presented here. The 2013 paper tried to suggest the potential for reconstructing discharge and one of the reviewers believes that this is not warranted at all because no experiments like in this study were performed. See the HESS discussion threads at: https://www.hydrol-earth-syst-sci.net/17/4577/2013/hess-17-4577-2013-discussion.html. Consequently, all the related suggestions on flow reconstruction (including the concept of doing so) were removed from the Pan and Wood (2013) paper.

This study puts forward the concept of streamflow interpolation based on inverse streamflow routing for the first time, designs the actual workflow for producing these streamflow estimates,

and then conducts experiments to test and validate the interpolation framework. This concept and actual workflow was never framed, tested, or validated in the 2013 paper by Pan and Wood. We think such a step in this paper is critical, even though the main mathematical mechanism for propagating flow information backward in time and space was established in Pan and Wood (2003).

[revised manuscript text omitted]

---

## Referee Report (RR1)

[referee-annotated manuscript omitted]

---

## Author Response (AR2)

**Responses to Reviewer Comments**

Dear HESS editor:

We would like to begin by thanking you, Dr. Read, and the anonymous reviewer for your great help throughout the revision of this manuscript. The comments, criticisms, suggestions, insights and guidance provided to us throughout this process have been invaluable to improving our work and we believe that this manuscript has been greatly improved. We have addressed the major and minor issues raised and revised the manuscript accordingly.

The reviewer comments are quoted in normal **black** font, the responses are highlighted in **blue** color, and the text from the revised manuscript is cited in **red** color.

Sincerely,

Colby K. Fisher, Ming Pan, and Eric F. Wood

**Responses to Reviewer 2:**

Major comments:

1) The new method requires assumption of parameters: wave velocity and Kalman filter smoothing window. How uncertain are these compared to parameter assumptions for statistical methods that can achieve estimates of runoff at ungauged locations? There is a sentence or so on why the new method is 'better', but it is still unclear to me why someone would choose this new method over/instead of PUB methods.

With regards to the virtues of the ISR methodology versus the statistical methods used in PUB, we believe that one of the key differentiators is the fact that the ISR method accounts for a physical representation of the river system (in the form of a river routing mode, albeit a simple one), which is an important feature. We do agree with the reviewer that our method requires the assumption of parameters in basins where we may not know their most appropriate values, but it is our belief that this difference provides strong evidence that there should be better physical consistency than with a purely statistical method. In some cases the ISR method may prove to be worse than other statistical methods currently used with a known set of parameters. The authors of this study believe that further work should be done in future studies to test both types of models in a controlled set of simulations where the results can be easily compared. While valuable, such a comparison was beyond the scope of this manuscript.

2) Why was the new approach tested only using 1 year (2009)? If it is computationally efficient and USGS data is very easy to process, I am not sure why the case study is so limited to a single year, which in my opinion, limits the extrapolation potential for understanding how the method can reproduce a full distribution of flows.

The goal of this paper was to perform experiments to illustrate the feasibility of a numerical method to reproduce given discharge records. While the method is relatively computationally efficient and we agree that the USGS data is easy to access and process, this choice was made from the beginning of the study to limit our period of analysis to a short enough time period such that we could perform extensive and easily repeatable experiments as a proof of concept. In the prior round of revisions we noted that the "year 2009 was an "average" and "typical" year for the study area (Ohio basin) with a normal seasonal cycle and flow values that cover most of the high and low ranges of regional hydrology. There were no dramatic/extended droughts or floods in 2009." We believe that such a choice minimizes the impact of many compounding factors like the deficiencies of simple diffusive wave routing under extreme conditions (high flow and low flow). Also experiments and observations over a "typical" period can be more generalizable. This information was added into the text as part of the previous round of revisions lines 16-18 P6. We do acknowledge that additional experiments are necessary to test the proposed methodology under more "extreme" conditions to better illustrate the performance under a wide range of possible conditions for any given basin. Such work was beyond the scope of this proof of concept study but will be a subject of future work. Please see the following article currently in press:
Yang, Y., P. Lin, C. K. Fisher, M. Turmon, J. Hobbs, C. M. Emery, J. T. Reager, C. H. David, H. Lu, K. Yang, Y. Hong, E. F. Wood, and M. Pan, 2019: Enhancing SWOT Discharge Interpolation through Spatio-temporal Correlations. Remote Sensing of Environment, in press.

Minor Comments:

L 1, P 1: Authors should choose one of these words (Assimilation/Interpolation), whichever is most appropriate.

We thank the reviewer for their suggestion; however, we believe that this title best represents the core processes of the ISR methodology. While it is true that the experiments with and without initial guess can be seen as "data assimilation", such a distinction is very important for satellite remote sensing applications like the future SWOT mission because estimation with no initial guess will enable the use of "satellite-only" products, which is equivalent to performing "interpolation", instead of "data assimilation" (i.e. satellite-model combined products). For this reason, we prefer to keep the title as it is. This clarification has also been added to the text in lines 3-7 P 4:

"As such, this method can be seen as performing both data assimilation and interpolation. While it is true that the experiments with and without an initial guess can be seen as "data assimilation", such a distinction is very important for satellite remote sensing applications such as the future SWOT mission because estimation with no initial guess will enable the use of "satellite-only" products, which is equivalent to performing "interpolation", instead of "data assimilation" (i.e. satellite-model combined products)."

L 7, P 1: In my opinion water use is not a central topic of this paper, in fact its much broader (budgets and variability are primary topics).

Thank you for pointing this out. We have adjusted the abstract accordingly (L 6 P 1):
"Poorly monitored river flows in many regions of the world have been hindering our ability to accurately estimate global water budgets as well as the variability of the global water cycle."

L 15, P 2: How do statistical approaches to approximate/estimate flow at ungauged basins fit in?

Please see the response to the major comment (1) above.

L 14, P 3: I assume this clause refers to the new method, but it is unclear.

This clause refers to the other available methods, which have tended to directly use observations of water height (the state variable) to adjust the discharge of a given reach, as opposed to our methodology, which uses observations of discharge directly.

[revised manuscript text omitted]